# Neuropharmacological Potential of Diterpenoid Alkaloids

**DOI:** 10.3390/ph16050747

**Published:** 2023-05-14

**Authors:** Arash Salehi, Mustafa Ghanadian, Behzad Zolfaghari, Amir Reza Jassbi, Maryam Fattahian, Parham Reisi, Dezső Csupor, Ikhlas A. Khan, Zulfiqar Ali

**Affiliations:** 1Department of Pharmacognosy, School of Pharmacy and Pharmaceutical Sciences, Isfahan University of Medical Sciences, Isfahan 81746-73461, Iran; arashsalehi94@yahoo.com (A.S.); behzadz@gmail.com (B.Z.); fattahian.maryam@gmail.com (M.F.); 2Isfahan Pharmaceutical Sciences Research Center, School of Pharmacy and Pharmaceutical Sciences, Isfahan University of Medical Sciences, Isfahan 81746-73461, Iran; 3Medicinal and Natural Products Chemistry Research Center, Shiraz University of Medical Sciences, Shiraz 71348-14336, Iran; arjassbi@gmail.com; 4Department of Physiology, School of Medicine, Isfahan University of Medical Sciences, Isfahan 81745-33871, Iran; parhamzh@gmail.com; 5Institute of Clinical Pharmacy, Faculty of Pharmacy, University of Szeged, 6720 Szeged, Hungary; csupor.dezso@szte.hu; 6National Center for Natural Products Research, Research Institute of Pharmaceutical Sciences, School of Pharmacy, University of Mississippi, University, MS 38677, USA; ikhan@olemiss.edu

**Keywords:** diterpene alkaloids, Ranunculaceae, neuropharmacology, anticonvulsants, analgesics, acetylcholine receptors, dementia, antidepressive agents

## Abstract

This study provides a narrative review of diterpenoid alkaloids (DAs), a family of extremely important natural products found predominantly in some species of *Aconitum* and *Delphinium* (Ranunculaceae). DAs have long been a focus of research attention due to their numerous intricate structures and diverse biological activities, especially in the central nervous system (CNS). These alkaloids originate through the amination reaction of tetra or pentacyclic diterpenoids, which are classified into three categories and 46 types based on the number of carbon atoms in the backbone structure and structural differences. The main chemical characteristics of DAs are their heterocyclic systems containing β-aminoethanol, methylamine, or ethylamine functionality. Although the role of tertiary nitrogen in ring A and the polycyclic complex structure are of great importance in drug-receptor affinity, in silico studies have emphasized the role of certain sidechains in C13, C14, and C8. DAs showed antiepileptic effects in preclinical studies mostly through Na^+^ channels. Aconitine (**1**) and 3-acetyl aconitine (**2**) can desensitize Na^+^ channels after persistent activation. Lappaconitine (**3**), N-deacetyllapaconitine (**4**), 6-benzoylheteratisine (**5**), and 1-benzoylnapelline (**6**) deactivate these channels. Methyllycaconitine (**16**), mainly found in *Delphinium* species, possesses an extreme affinity for the binding sites of α7 nicotinic acetylcholine receptors (nAChR) and contributes to a wide range of neurologic functions and the release of neurotransmitters. Several DAs such as bulleyaconitine A (**17**), (**3**), and mesaconitine (**8**) from *Aconitum* species have a drastic analgesic effect. Among them, compound **17** has been used in China for decades. Their effect is explained by increasing the release of dynorphin A, activating the inhibitory noradrenergic neurons in the β-adrenergic system, and preventing the transmission of pain messages by inactivating the Na^+^ channels that have been stressed. Acetylcholinesterase inhibitory, neuroprotective, antidepressant, and anxiolytic activities are other CNS effects that have been investigated for certain DAs. However, despite various CNS effects, recent advances in developing new drugs from DAs were insignificant due to their neurotoxicity.

## 1. Introduction

Diterpenoid alkaloids (DAs) are characteristic components of some genera of the Ranunculaceae family, the occurrence of which is extraordinarily high in the genera *Aconitum*, *Delphinium*, and *Consolida*. Almost half of these natural compounds have been isolated from *Aconitum* sp. [1,2,3,4,5]. Despite being highly toxic, DAs possess interesting complex structures and noteworthy pharmacological and physiological effects which have attracted an exponentially increasing interest worldwide. Special emphasis has recently been placed on the research activities concerning the pharmacological and toxicological evaluations of these alkaloids. DAs are derivatives of the two nitrogen-free diterpenoids, *ent*-kaurene and *ent*-atisane, and therefore may be considered pseudoalkaloids. These compounds receive their nitrogen atoms from amino acids through an enzymatic transamination by aminotransferases [6]. *Aconitum* species commonly known as monkshood or wolfsbane were regarded as extremely toxic plants anciently used as arrow poisons, whereas *Delphinium* species have been considered to be less toxic [7]. Although DAs are known for their neurotoxicity, they exhibit a broad range of promising biological effects including analgesic, anticonvulsant, neuroprotective antiarrhythmic, antitumor, antimicrobial, and local anesthetic activities [8,9,10,11]. Before 1821, the *Consolida* species were classified in the *Delphinium* genus as a phylogroup. Since these two genera are morphologically very similar, they are repeatedly confused [12]. Regarding the similar chemical composition, their medicinal application has overlapped. *Consolida* sp. has been used for hundreds of years in China and the west of Asia as a medicinal plant for several nociceptive conditions, e.g., rheumatism, sciatica, and also as an anthelmintic [13,14]. Apart from the Ranunculaceae plants, nine species from *Spiraea* (Rosaceae) were found to have DAs [15]. The medical application of *Spiraea* sp. refers to traditional Chinese medicine which is used in some inflammatory and painful conditions such as toothache and headache. In addition, some sporadic studies reported the presence of DAs in *Thalictrum* (Ranunculaceae), *Inula* (Asteraceae), *Garrya* (Garryaceae), *Erythrophleum* (Fabaceae), *Anopterus* (Escalloniaceae), *Rumex Pictus* (Polygonaceae), *Caesalpinia Sappan* (Fabaceae), *Artemisia korshinskyi* (Asteraceae), and *Isodon rubescens* (Lamiaceae) species [6,15]. In traditional Chinese and Japanese medicine, for centuries, plants containing DAs have conventionally been used after being processed into less toxic products [16]. The unique pharmacological impacts of DAs on the central nervous system (CNS) encouraged scientists to seek new lead compounds (Figure 1). Several authors have previously reviewed the recent evidence associated with DAs’ diversity, toxicity, biosynthesis, pharmacokinetics, structure–activity (or toxicity) relationship, and their analgesic, cytotoxicity, anti-inflammatory, and antimicrobial activities. However, the anticonvulsant, antagonizing α7 nAChR, antidementia, and neuroprotective effects as well as the biomolecular mode of action have not been concluded [6,8,17]. This narrative review aimed to summarize the CNS effects and biomolecular mechanism of action of DAs and plants containing DAs, along with classification, structural diversity, biosynthetic pathway, and the aspects restricting their widespread utilization mainly from the *Aconitum* and *Delphinium* species. A systematic search was conducted through the databases PubMed, Scopus, and Web of Science in December 2021.

## 2. Chemistry and Biosynthetic Pathway

### 2.1. Chemical Classification

Diterpenoid alkaloids originate through the amination reaction of tetra or pentacyclic diterpenoids. Based on the number of carbon atoms in the backbone structure, the 46 identified types are classified into three categories: C18, C19, and C20 diterpenoid alkaloids. These alkaloids possess a heterocyclic system containing β-aminoethanol, methylamine, or ethylamine functionality [4].

The small category of C18-DAs formed by demethylation at the C4 position of C19-DAs. C18-DAs are divided into five types: ranaconitines, lappaconitines, puberudines, puberunines, and sinomontadines (Figure 1). Lappaconitines possess a methine at C7 while ranaconitines are characterized by an oxygen-containing group at this position [8,18]. The second difference is in the C4 position, where the ranaconitines have a methine unit, while lappaconitines contain a methine unit or an oxygenated quaternary carbon (several compounds have been reported to possess chloro-substituents at C4). The three remaining types have undergone rearrangements as puberudines (open A ring), puberunines (unusual E ring), and sinomontadines (seven-membered A ring) [6].

It was reported that napeline and denudatine type of C20-DAs could lose a carbon unit to originate C19-DAs by the action of specific cyclases [6]. C19-DAs are structurally classified into six main types, aconitine, lactone, lycoctonine, 7,17-seco, franchetine, and glycosides, to gather with six rearranged subtypes which are chemically biosynthesized from aconitine type: acoseptines, vilmoraconitines, hemsleyaconitines, vilmotenitines, grandiflodines, and N-formyl-4, 19-secopacinine (Figure 2). Glycosides are characterized by a sugar unit (L-Ara*p* and L-Ara*f*) at the C1 or C14 positions of the aconitine structure [6]. C19-DAs are the largest category and hundreds of them were isolated from *Aconitum* and *Delphinium* species [6,8].

C20-DAs have a more complex structure which is characterized by a tetracyclic diterpenic backbone. *Delphinium* species are the major source of this category. C20-DAs are divided into 23 types: atisine, denudatine, hetisine, hetidine, napelline, vacognavine, veatchine, and anopterine, besides fifteen rearranged types. Atisine is assumed as the original group with a pentacyclic core structure and nitrogen which is located between C19 and C20. Denudatine is hexacyclic by an additional bond between C7 and C20. Hetisine is characterized by two excess bonds (N-C6 and C14-C2) in atisine. Hetidine structure is achieved by a connection between C14 and C20 in hetisines. The hexacyclic backbone of vokognavine is achieved by breaking the hetisine bond at N-C19 (forming seco). The pentacyclic backbone of veatchine is similar to kaurane diterpenes. Napelline and anopterine are similar to veatchines with the additional C7-C20 and C14-C20 bonds, respectively. There are also fifteen rearranged types in C20-DAs: spireine, cardionidine, albovionitine, arcutine, aconicarmisulfonine, kaurine A, kaurine B, delnudine, kusnesoline, grandiflodine, tricalysiamide, caesanine, anthriscifolsine, actaline, and racemulosine (Figure 3) [6,8].

### 2.2. Biosynthetic Pathway

The five-carbon unit, isoprenoid (IPP), which is formed through methylerythritol (MEP) or mevalonate (MVA) pathways, initiates the biosynthesis of terpenoids [19]. Four IPP units bind (catalyzed by geranylgeranyl pyrophosphate synthase) to form a 20-carbon geranylgeranyl pyrophosphate (GGPP) which further cyclizes by an enzymatic reaction to produce *ent*-copalyl diphosphate (*ent*-CPP) [20,21]. Among all the identified diterpenoid skeletons, only the two *ent*-kaurene and *ent*-atisane types can contribute to DA biosynthesis. These two precursors of DAs originate from *ent*-CPP by the actions of *ent*-kaurene/*ent*-atisane synthase [22]. The *ent*-kaurene transforms to provide the *ent*-kaurenoic aldehyde through two repeated oxidations by the *ent*-kaurene oxidase (a cytochrome P450 mono-oxygenases) [23]. On the other hand, L-serine is a plausible amino acid that could provide nitrogen for alkaloid biosynthesis [22]. Transferring nitrogen from L-serine to *ent*-kaurenoic aldehyde (an intermediate product) takes place through the enzymatic transamination by aminotransferases (serine-pyruvate transaminase and serine-glyoxylate transaminase are considered to have the major roles) (Figure 2).

Expressing *ent*-kaurene oxidase genes in plants is the reason for the presence of these alkaloids. Since various aldehydes may be produced in the plants, the position of nitrogen in DAs might be different [6]. It has been shown that C18 and C19-DAs biochemically derive from C20-DAs and the involved biochemical reactions are thought to be Wagner–Meerwein rearrangement, Prins cyclization (ajaconine subtype into hetidine class), Schiff base cyclization (atisine class to lycoctoine class), and Mannich reaction, which supports the hypothesis that cyclases have a huge impact on DAs diversity [24,25]. Monooxygenases of cytochrome P450 could prompt oxidative rearrangements and oxygenize various carbon positions on DAs’ structure [6,26]. Thus, the created hydroxyls are the site of action for methyltransferases and acyltransferases to produce more toxic methyl, benzoyl (Bz), acetyl (Ac), anisoyl (As), Cinnamoyl (Cn), Veratroyl (Vr), and ***O***-lipo derivatives [27].

## 3. Pharmacological Activities

To affect the central nervous system, primarily, the drug should pass the blood–brain barrier (BBB). Transmembrane diffusion is the most common route of drugs to pass the BBB and, in contrast to the transport system, shows a non-saturable kinetic. Physiochemical features of the drug mainly determine the amounts of this passage. Molecular weight (400–600 Da is optimum), lipid solubility, molecular charge, and tertiary structure are the most important factors necessary for transmembrane diffusion through BBB [28]. The diterpenic backbone of DAs provides their suitable lipid solubility, but the presence of tertiary nitrogen makes them different from normal diterpenoids. The tertiary nitrogen with the highest proton affinity in the molecule (in water) rearranges the electronic structure of DA by its protonation. Moreover, computational modeling showed that the function of nitrogen besides ester sidechains causes DAs to interact with the active sites as well as their toxicity [29,30].

### 3.1. Anticonvulsant Effects

An abnormal and transient neuronal synchronization in the brain causes epilepsy. This phenomenon alters the correct pattern of neuronal connections and is characterized by the peaks and troughs of electric discharge in the electroencephalograph (EEG) and may lead to various mental and physical symptoms associated with the origin of the disruption [31]. Voltage-gated sodium channels (VGSCs) play an essential role in the generation and coordination of action potential in the central nervous system, which participates fundamentally in the pathophysiology of seizures; thus, these channels could be an important aim for pharmaco-therapy [32]. The Fuzi total alkaloids (Aconiti Lateralis Radix Praeparata), an essential medicine in traditional Chinese medicine, significantly prolong the seizure latency time and decrease the mortality in pentylenetetrazole-induced mice [33]. Additionally, anticonvulsant effects of several alkaloidal-rich fractions and also isolated DAs from *Delphinium* and *Aconitum* species have been investigated (Table 1; Figure 4). Ameri’s findings have greatly enhanced our knowledge of the anticonvulsant effects of DAs [9]. Five distinct toxin binding sites were explored on voltage-dependent Na^+^ channels. Aconitine (**1**), batrachotoxin, and veratridine are tightly bound to site 2 in the α subunit of Na^+^ channels which are localized in the transmembrane region [34,35,36]. The voltage-dependent Na^+^ channels are activated by **1**, resulting in the depolarization of the presynaptic membrane [37]. Compound **1** activates Na^+^ channels at resting membrane potential permanently by blocking Na^+^ channel inactivation and causing hyperpolarization of threshold activation due to sustained Na^+^ influx; thereby, inexcitability occurs [9]. The structurally related 3-acetylaconitine (**2**) was shown to activate voltage-dependent Na^+^ channels by a similar mechanism of action [38]. In contrast to compounds **1** and **2**, some other *Aconitum* DAs exist, revealing inhibitory effects on the Na^+^ channels; lappaconitine (**3**), N-desacetyllappaconitine (**4**), 6-benzoylheteratisine (**5**), and 1-benzoylnapelline (**6**) reduce the peak amplitude of the current and show an inhibitory effect on the Na^+^ channel [9]. While **3** could antagonize the suppressive effect of 1 on population spike in rat hippocampal slices which are attributed to their opposite mode of action; both **1** and **3** possess an anticonvulsant potential [34,39,40]. Compounds **1** and **2** indirectly suppressed the uptake of [3H] noradrenaline via increasing sodium concentration in synaptosomes, while **3** and **4** do not indicate such an impact on [3H] noradrenaline uptake [9,41]. Despite Ameri’s investigations, Voss et al. showed acute seizure-like activity induced by **1** [42]. Songorine (**7**) might antagonize the GABA receptor in the rat brain via a different site of action different from the GABA recognition site [43]. However, in an in vivo experiment, **7** acted as an agonist on D2 and GABA A receptors, which might be translated as a potential antiepileptic effect [44]. The involvement of α-adrenoceptors and modulation of GABA receptors were reported for the antiepileptic activity of mesaconitine (**8**) and Fuzi total alkaloid, respectively [33,45].

### 3.2. Antagonizing α7 Nicotinic Acetylcholine Receptor

Nicotinic acetylcholine receptors (nAChRs) are cation-conducting ligand-gated receptors broadly expressed in the peripheral and central nervous systems and also in muscular cells. Seventeen nAChR subunits, comprising ten α, four β, γ, δ, and ε subunits have so far been identified in mammals where α7 homomeric nAChRs and heteromeric α4β2* nAChRs are predominant and expressed functionally in the brain. These receptors contribute to a range of neuronal functions from cognitive performance to neurotransmitter release (including glutamate, GABA, and dopamine) through altering intracellular Ca^2+^ concentration. It has been clarified that the pathogenesis of several neurological disorders containing Alzheimer’s disease, schizophrenia, Parkinson’s disease, and depression as well as nicotine addiction are related to α7 and α4β2* receptor functions. Although many efforts have been devoted to developing a (partial) agonist of α7 nAChR targeting neurologic and psychiatric conditions, none has already been approved as medicine due to their undesirable side effects [62,63,64]. Moreover, results from a meta-analysis indicated that curing the cognitive dysfunction in schizophrenia and Alzheimer’s disease with agonists of α7 nAChR cannot be considered a robust treatment [65]. In 1986, Jennings et al. found that the insecticidal effect of *Delphinium* plants is mainly due to the presence of methyllycaconitine (**16**) (Figure 5a), showing a high affinity to cholinergic receptors of insects [66]. Indeed, **16** is a potent and highly selective α7 nAChR antagonist which acts similarly to the α-bungarotoxin protein of snake venom (Figure 5b) [67]. Despite former research, sporadic studies have recently focused on the beneficial role which low doses of **16** play in neurologic disorders (Table 2); equivalent results were also reported for other nAChR antagonists, e.g., mecamylamine [68]. Several hypotheses have been proposed to explain their mechanism of action.

Based on the allosteric theory of ion channel functioning, nAChRs might have three different conformational states, resting, active, and desensitized, having the capability of altering their conformational states through spontaneous transitions. Ligand affinity highly depends on conformational states. On the other hand, α7 nAChRs possess five binding sites for interacting with agonists, where binding to two of them is necessary for activating the receptor and opening the channel. Binding an antagonist, e.g., **16**, to remaining binding sites may prevent or delay the desensitization of the receptor and affect the affinity of the receptor to other ligands. As a further explanation, the stoichiometry of the binding site with which the acetylcholine molecules (AChs) interact determines the degree of channel activation. The binding of two AChs to two consecutive binding sites leads to slow channel activation, while binding to two non-consecutive sites results in rapid activation and receptor desensitization. The binding of **16** impacts the binding pattern of ACh to sites. In case one, two, or three molecules of **16** connect to the binding sites, the probability of non-consecutive binding of ACh increases, whereas binding four and five molecules of **16** eliminates the possibility of simultaneous binding of two ACh molecules, which is essential for the activation of the receptor. The hypothesis explains the contradictory responses of α7 nAChR to lower and higher doses of **16** (Figure 6) [64].

### 3.3. Analgesic Activity

Nociceptive pain develops in response to specific noxious stimulation of normal tissue in a normal somatosensory nervous system; in contrast, neuropathic pain comes from damage to the nerves or nervous system; there is also a third kind of pain called inflammatory pain, which may be described as increased sensitivity due to the inflammatory response associated with tissue damage [73]. Herbal preparations containing *Aconitum* have been widely used in traditional Chinese medicine to relieve rheumatoid arthritis [74]. The effectiveness of *Aconitum*-containing preparations for diabetic peripheral neuropathic pain was shown in a small clinical trial [75]. Pharmacological in vivo studies confirmed the antinociceptive activity of various DAs and extracts from *Aconitum* species (Table 3; Figure 7). Bulleyaconitine A (**17**), a diterpenoid alkaloid isolated from the rhizomes of *A. bulleyanum*, is structurally related to **1** and was approved in 1985 in China for the alleviation of chronic pain and is available as tablets, intramuscular injections, and soft gel capsules [76]. Compound **17** showed more significant analgesic potency than morphine. The clinical utility of **17** in the three past decades in China has demonstrated the advantages of its application for relieving chronic pain without any serious side effects, contrary to what is observed in the case of the application of morphine or non-steroidal analgesics [77]. Several studies have emphasized the positive central effect of **17** in the treatment of nociceptive pains [63]. The analgesic effect of **17** could be antagonized by reserpine but not naloxone, which suggests an involvement of catecholamine in the **17** mechanisms of action [78]. Compound **7** revealed a visceral antinociceptive effect by stimulating the release of dynorphin A, leading to the activation of presynaptic κ-opioid receptors in afferent neurons [66]. Consistently, it was indicated that **17** can directly stimulate the dynorphin A expression in spinal microglia [79]. On the other hand, Xie et al. suggested a preferable blocking effect on tetrodotoxin-sensitive Nav1.7 and Nav1.3 in dorsal root ganglion neurons as the mode of action of **17** [80]. Compound **17** could also inhibit Nav1.7 and Nav1.8 Na^+^ currents in a use-dependent manner [81]. Depressing of long-term potentiation at C-fiber synapses in the spinal dorsal horn and inhibition of transmitter release in paclitaxel-induced neuropathic pain were accounted for as well [82].

Another diterpenoid alkaloid, lappaconitine (**3**) from *Aconitum* species revealed antinociceptive effects in nociceptive test models. Compound **3** has been used for its analgesic properties in both traditional Chinese and Kampo medicine [83]. Its analgesic potency is comparable to that of tramadol and morphine [84,85]. Pretreatment with antagonists of adrenergic and serotoninergic systems could decrease the antinociceptive effects of **3**, which might be mediated by β-adrenoceptors and 5-HT2 receptors in the brain and α-adrenoceptors and 5-HT receptors in the spinal cord [86]. Moreover, when **3** was administered via subcutaneous or intracerebroventricular routes, it could inhibit behavioral response induced by substance P or somatostatin, while the intrathecal injection showed no effect. These observations provide some evidence to support the supra-spinal descending mechanism of action of **3** and suppressing the transmission of the nociceptive neuronal message [87]. Inhibition of P2X(4) expression in microglia in the dorsal horn and P2X3 receptors in dorsal root ganglia neurons in the spinal cord were demonstrated as further plausible mechanisms of action [88,89,90]. Li et al. indicated that inhibition of the Nav1.7 channel is involved as well [91]. Similar to **17** stimulation of the expression of spinal microglial dynorphin A was suggested [92]. Ono et al. indicated that the antinociceptive effect of **3** was not antagonized by naloxone in the hot plate and acetic acid-induced writing tests, contrary to what was observed in the tail pinch test. It was suggested that the action of **3** was only partially antagonized by naloxone, which explains the naloxone-resistant antinociceptive activity for **3** at the spinal sites [93].

Mesaconitine (**8**), one of the most important ingredients of *Aconitum* species, also exhibits antinociceptive activity in animal models. Subcutaneous administration of **8** has shown much greater analgesic activity compared to that of morphine in animal models [94]. Friese et al. categorized *Aconitum* alkaloids into two groups based on their binding affinity to Na^+^ channel epitope site 2. Alkaloids of the high-affinity group (**1** and **8**) are supposed to act as Na^+^ activators due to their increasing synaptosomal [Na^+^]i and [Ca^2+^]i effect. A higher antinociceptive action was shown in the case of the high-affinity group than the other, indicating the importance of Na^+^ channels in the activity of **8** [95]. The analgesic action of **8** was not inhibited by levallorphan (an opiate antagonist) in an animal model. On the other hand, **8** could increase the release of endogenous norepinephrine from sympathetic nerve fibers [94]. Application cyclic AMP and isoproterenol (a β receptor agonist) in combination with **8** enhanced its analgesic activity, while propranolol (a β receptor antagonist) suppressed it, which magnifies the role of the adrenergic system in the mode of action [96]. Involvement of the nucleus reticularis gigantocellularis (NRGC), the nucleus reticularis paragigantocellularis (NRPG), the periaqueductal gray (PAG), and the lumbar enlargement in the antinociceptive action of **8** was shown in a rat model. Indeed, **8** might activate the inhibitory noradrenergic system in descending neurons from the NRPG, especially through β-receptor-mediated effects of noradrenaline [86,87]. Moreover, it was also revealed that the nucleus raphe magnus (NRM) has a key role via the serotoninergic system in the mechanism of action [97].

Isotalatizidine (**18**) attenuates allodynia in neuropathic pain in a mice model. An increase in the expression of dynorphin A was shown, and its release from microglia was observed, which might be triggered by the activation of the ERK1/2 pathway and phosphorylation of CREB (cAMP response element-binding) [98].

Bullatine A (**19**) was shown to have a synergistic effect with morphine as it increases morphine analgesic action while counteracting morphine tolerance. Stimulation of spinal microglia and subsequent enhancing of the expression of dynorphin A were thought to have a role [99,100].

**Table 3 pharmaceuticals-16-00747-t003:** Antinociceptive effects of fractions/constituents from *Aconitum* and *Delphinium* species.

Plant	Used Part/Constituents	Biological Activity	Ref.
*Aconitum episcopale*	Episcopaline B (**20**)	Antinociceptive effect 2-fold lower than aspirin and acetaminophen	[101]
*Aconitum pseudostapfianum*	Pseudostapine C (**21**)	2-fold more potent antinociceptive effect than aspirin and acetaminophen	[102]
*Aconitum episcopale*	Episcopine A (**22**)	2-fold more potent antinociceptive effect than aspirin and acetaminophen	[103]
*Aconitum* *carmichaelii*	18	Antinociceptive effect	[98]
*Aconitum carmichaeli*	Plant extract and neoline (**23**)	Attenuated the mechanical hyperalgesia	[104]
*Aconitum carmichaeli*	Plant extract and **23**	Attenuated cold and mechanical hyperalgesia	[105]
*Aconitum carmichaeli*	Aqueous extracts	Antinociceptive activity	[106]
*Aconitum carmichaeli*	Processed *aconitum* tuber	High doses of processed Aconiti tuber inhibit the acute but potentiate the chronic antinociception of morphine	[107]
*Delphinium denudatum*	Aqueous root extract	Antinociceptive activity	[108]
*Aconitum carmichaeli*	Processed *aconitum* tuber and **8**	Antinociceptive activity of 8 was more potent than morphine	[109]
*Aconitum* sp.	1	Significant analgesic effects	[110]
*Aconitum carmichaelii*	Aconicatisulfonines A (**24**) and B (**25**)	Analgesic activities	[111]
*Delphinium denudatum*	Ethanolic extract and methanol fraction	Analgesic activity	[112]
*Aconitum kusnezoffii*	**23**	Analgesic activity	[113]
*Aconitum carmichaelii*	Aconicarmichosides E (**26**), F (**27**), H (**28**), I (**29**), and J (**30**)	Analgesic activity	[114]
*Aconitum* sp.	**3**	Relieves the pain	[82]
*Aconitum baikalensis*	Napelline (**31**), hypaconitine (**32**), **7**, **8**, 12-epinapelline *N*-oxide (**33**).	Analgesic activity comparable to that of sodium metamizole	[115]
*Aconitum* *weixiense*	Weisaconitines D (**34**)	Analgesic activity	[116]
*Aconitum* *carmichaeli*	Guiwuline (**35**)	Potential analgesic activity	[117]
*Aconitum* *carmichaeli*	8-*O*-cinnamoylneoline (**36**)	Analgesic activity	[118]

### 3.4. Antidementia Effect

A decrease in acetylcholine (ACh) in the central nervous system is associated with dementia. Several DAs possess an acetylcholinesterase (AChE) inhibitory effect (Table 4; Figure 8), and therefore, these compounds may theoretically act as antidementia agents. Some evidence has emphasized the role of K^+^ channel blockers as drug candidates for the treatment of neurodegenerative disorders [119]. Virtual screening from the Chinese natural product database resulted in the identification of four *Aconitum* alkaloids, namely, songorine (**7**), pyrochasmaconitine (**37**), 14-benzoyl talatisamine (**12**), and talatisamine (**13**), as candidates for this effect; an electrophysiological assay of rat dissociated hippocampal neurons verified this assumption [120]. All of the four alkaloids potently inhibited the delayed rectifier K^+^ current, while **13** was slightly effective on Na^+^ and Ca^2+^ channels in rat hippocampal neurons [119].

Macroautophagy was previously considered one of the plausible mechanisms of the pathogenesis of Alzheimer’s disease (AD). Compound **16** protects the human neuroblastoma SH-SY5Y cell line against amyloid-β peptide-induced cytotoxicity. Compound **16** might prevent autophagy induced by amyloid-β through the mammalian target of the rapamycin (m-tor) pathway [121]. Apetalrine B (**38**), the semisynthetic DAs from its parent compound, aconorine (*A. apetalum*), exhibits a neuroprotective effect in H_2_O_2_-treated SH-SY5Y cells by inhibiting cell apoptosis [122]. Firstly, **7** was superior to piracetam (a nootropic agent) in the correction of the cholinergic abnormalities induced by scopolamine, which indicates its antiamnestic activity. Secondly, **7** might act as a cholinomimetic, antihypoxic, and cerebro-protective agent [123,124]. Among the DAs from *A. anthoroideum*, rotundifosine F (**39**) possesses AChE inhibitory activity. Moreover, nominine (**40**) represented a neuroprotective effect due to its protection against 1-methyl-4-phenylpyridinium (MPP+)-induced apoptosis in SH-SY5Y cells [10]. Fuzi considerably induced growth of cell projection and cell proliferation in 0.4–0.8 mg/mL and 1.6–100 mg/mL, respectively, in the AD cell model. In addition, the involvement of GRIN1 and MAPK1 genes contributed to the antiAD mechanism of action [125].

**Table 4 pharmaceuticals-16-00747-t004:** Antidementia effect of fractions/constituents from *Aconitum* and *Delphinium* species.

Plant	Used Part/Constituents	Biological Activity	Ref.
*Aconitum hemsleyanum*	Hemsleyaline (**41**)	Mild AChE inhibitory effect	[126]
*Aconitum kirinense*	Diterpenoid alkaloids	Moderate AChE inhibitory effect	[127]
*Aconitum laeve*	Diterpenoid alkaloids	Swatinine-C (**42**) and hohenackerine (**43**) competitively inhibited AChE and BChE Aconorine (**44**) and **3** noncompetitively inhibited AChE	[128]
*Delphinium denudatum*	Diterpenoid alkaloids	Jadwarine-A (**45**), **18**, and dihydropentagynine (**46**) competitively inhibited AChE and BChE, while 1β-hydroxy,14β-acetyl condelphine (**47**) and jadwarine-B (**48**) showed non-competitive inhibition	[129]
*Aconitum heterophyllum*	Diterpenoid alkaloids	Compounds 6b-methoxy, 9b-dihydroxylheteratisine (**49**), 6,15b-dihydroxylhetisine (**50**), iso-atisine (**51**), heteratisine, 19-epi-isoatisine (**52**), and atidine (**53**) non-competitively inhibited AChE and BChE, while compounds 1a,11,13b-trihydroxylhetisine (**54**) and hetisinone (**55**) were determined as competitive inhibitors	[130]
*Delphinium denudatum*	Isotalatizidine (**18**) hydrate	Potent dual cholinesterase inhibitor	[131]
*Aconitum heterophyllum*	Heterophyllinine A(**56**) and B (**57**)	**56** and **57** inhibited AChE and BChE enzymes	[132]
*Aconitum falconeri*	Faleoconitine (**58**) and Pseudaconitine (**59**)	Moderate inhibitory activity on AChE	[133]

### 3.5. Antidepressant Effects

Two antagonists of nicotinic acetylcholine receptor (nAChR) were demonstrated to induce antidepressant effects in the forced swim and tail suspension tests in mice, while agonists of nAChR did exhibit no similar activity. The observed effect was more potent in the case of the non-selective nAChR antagonist mecamylamine compared to that of **16** (an α7 nAChR selective antagonist) [134]. Modulating serotonin sensitivity is considered a promising mechanism of action of DAs’ antidepressant activity in *Aconitum baicalense*. Besides their positive results in the tail suspension test, these compounds showed an antiexudative effect in edema induced by serotonin in mice. These alkaloids were **7**, **8**, **32**, and **31** in order from most to least potent [135]. Antidepressant actions were also reported for chronic administration of Fuzi total alkaloids in normal and ovariectomized mice. An study designed to elucidate the mode of action discovered the potential roles of cAMP response element-binding (CREB) and brain-derived neurotrophic factor (BDNF) protein pathways [136]. Moreover, the sex-dependent antidepressant effect of **16** was consistently reported in male mice, while female mice did not exhibit the same. Local administration of **16** in male mice hippocampus could reverse the depression-like effect which was induced by physostigmine [137].

### 3.6. Miscellaneous Effects

Compound **7** has exhibited a more potent anxiolytic activity compared to phenazepam applying the Vogel conflict test in mice without a sedative effect. Anxiolytic activity of **7** was noted in a dose of 0.25 mg/kg, while a 10 times higher dose (2.5 mg/kg) showed an inferior activity [138].

An ethanolic extract of *Delphinium denudatum* has protected the brain against injury in a rat model of Parkinson’s disease. The expression of ipsilateral tyrosine hydroxylase was increased. The extract also prevented lipid peroxidation, depleted reduced glutathione in the substantia nigra, and attenuated the activity of superoxide dismutase and catalase in the striatum. Reduction in dopamine concentration was also reported in the striatum after the injection of 6-hydroxydopamine [139].

## 4. Clinical Trials

A double-blind, randomized, placebo-controlled, parallel-group, multi-center study was conducted on seventy-five patients with postherpetic neuralgia to evaluate the analgesic activity of bulleyaconitine A. The intervention group received tablets of bulleyaconitine A besides the first-line therapy (gabapentin), while the control group took gabapentin and placebo tablets. Treatment with bulleyaconitine A significantly improved the effective rate and the time taken to reach the primary outcomes in comparison with the control group [140]. In a clinical study, post-operative analgesia in one hundred and twenty participants was assessed for lappaconitine in four treatment groups: A: lappaconitine (12 mg) + bupivacaine (22.5 mg); B: lappaconitine (12 mg); C: bupivacaine (22 mg); D: morphine (2 mg) by epidural injection. The order of efficacy was D > A > C > B, while the onset of action was lower in the case of groups A and C [141]. Intravenous injection of lappaconitine was explored in a controlled clinical study for relief of pain after rectum carcinoma surgery. Sixty patients were divided into three groups to receive intravenous A: lappaconitine (8 mg), B: tramadol (100 mg), and C: Pethidine (50 mg) + droperidol (2.5 mg). Pain assessment was carried out using a visual analog scale (VAS) at different intervals. Lappaconitine significantly decreased the pain in patients who had undergone surgery for rectum carcinoma. Its VAS score was less than group D, while there was no significant difference between intravenous injection of 8 mg lappaconitine and 100 mg tramadol for pain relief after surgery [85]. In another study, of fifty-six patients after cholecystecystectomy, 40 mg lappaconitine showed an equal analgesic effect to 0.4 mg fentanyl, but with **less** nausea, vomiting, and itching [142].

## 5. Discussion

Although *Aconitum* and *Delphinium* species have been used for millennia in traditional medicine in Eastern societies, to date, there have been insufficient clinical studies evidencing their safety [74,83]. These plants contain DAs with characteristic bioactivities, and the use of these compounds, particularly in pure form, lends a fresh perspective to medicine. Despite the noticeable bioactivities of DAs that might be beneficial in various diseases, these compounds have played no major role in drug development in recent years. This gap is justified by the fact that there is a narrow therapeutic window between effective and toxic doses of *Aconitum* plants and their constituents.

Four active pharmacophores have been described previously for **1**: a tertiary nitrogen atom, a hydroxyl at C13, a benzoyl ester at C14, and acetyl at C8, which contribute to the Na^+^ channel-activating effect. DAs with Na^+^ channel-blocking effects have a similar backbone structure to **1** but it was explained that lacking each of the four active pharmacophores causes them to exhibit contrasting effects. Accordingly, it was also indicated experimentally that **3** could reverse the arrhythmogenic effects of **1** [30,143,144]. A more recent in silico study of an electronic structure investigation of 19 DAs revealed that what makes the difference is not different structures but a wide distance between their efficiency, i.e., the blockers ban the channel conductance, while the activators permit only a very slow ion flow. In addition, the computational analysis demonstrated that both activators and blockers could interact with the binding site of Na^+^ channels via the benzoyl ester residue [30].

The presence of a tertiary N-ethyl substituted in ring A and a saturated ring D, in addition to an OAc or OC_2_H_5_ moiety at C8 and an aromatic ester (O-Bz or O-As) at C14, which is necessary for analgesic activity. A 3α or 5-hydroxyl group may support the analgesic effects but methylation of C1 hydroxyl will decrease the potency [6,145].

The lycoctonine backbone is the base for interfering cholinergic receptors which are responsible for *Delphinium* intoxication. A methylsuccinylanthranoyl ester at C18 (e.g., compound **16**) is necessary due to providing proper orientation and improving binding affinity. The C14 functionality and the tertiary nitrogen also affect the activity [146].

The compounds responsible for the toxic effects are diester alkaloids which contain an acetyl group in C-8 besides a benzoyl or anisoyl functionality in C-14 [6,74,147]. A QSAR analysis investigated the structural toxicity relationship of diterpenoid alkaloids explaining that the number of C=O groups, lipophilicity, the atomic packing density in the molecular volume, surface polarity, and lipophilicity contribute to DA toxicity [29]. The most toxic compound in aconite is **1** with a lethal dose of 2 mg for an adult [148]. The main poisoning mechanism of action of **1** is the deactivation of the voltage-dependent sodium channels which leads to the disruption of neuromuscular release [9]. The cardiovascular system is another site where aconite derivatives can cause lethal ventricular arrhythmia [6]. Compound **16** exerts its toxicity through a different mechanism, i.e., by reversibly blocking the action of nicotinic acetylcholine receptors (nAChR), thereby paralyzing the respiratory muscles [149].

Several methods were emphasized to reduce the toxic effects of aconites in traditional Chinese medicine substantially based on heating processes (boiling, cooking, baking, steaming, and autoclaving), resulting in the hydrolysis of diester DAs to the less toxic monoesters and lipoalkaloids [148,150]. However, the medical effects do not necessarily decrease during the processing, as was demonstrated in the example of the anti-inflammatory activities of native DAs and their lipoalkaloid derivatives [151]. Additionally, the concomitant use of plants such as ginseng, licorice, dried ginger, and dahuang with aconite is believed to reduce its toxic effects and enhance its efficacy in traditional Chinese medicine [148].

## 6. Conclusions

As a conclusion to the above literature review, DAs originate from ent-atisane and ent-kaurene diterpenes through an enzymatic amination reaction. To date, 46 types of DA have been identified from three main categories: C18, C19, and C20-diterpenoid alkaloids. One of their most obvious effects is the anticonvulsant activity caused by interfering with the voltage-gated Na^+^ channels in the central nervous system through desensitization after persistent activation (e.g., **1** and **2**) or deactivation (e.g., **3**–**6**) of Na^+^ channels. The most effective chemical moieties involved in the binding of DAs to Na^+^ channels are four pharmacophores: the tertiary nitrogen atom, hydroxyl at C13, benzoyl ester at C14, and acetyl at C8. Some of these alkaloids showed promising analgesic effects. For example, **3**, **8**, **17**, and **18** exert potent activity by increasing the release of dynorphin A, activating the inhibitory noradrenergic neurons in the β-adrenergic system, or preventing the transmission of pain messages by inactivating Na^+^ channels. On the other hand, **16** has mixed effects. It is marked as a potent antagonist of α7 nACh receptors while reversely activating it in very low concentrations. The tertiary N-ethyl substituted in ring A, a saturated ring D, an OAc or OC**_2_**H**_5_** moiety at C8, and an aromatic ester (OBz or OAs) at C14 are likely to be necessary for analgesic activity. Although the role of tertiary nitrogen in ring A and polycyclic structure is of great importance in drug-receptor affinity, in silico studies have emphasized the role of sidechains in C13, C14, C8, etc. Moreover, some of the DAs have potent acetylcholinesterase inhibitory activity, which indicates their potential against neurodegenerative disorders such as Alzheimer’s disease. Antidepressant and anxiolytic effects have also been investigated in certain surveys; nevertheless, more experiments should be conducted to provide clearer evidence. Altogether, due to the undesirable profile of toxicity, no DA has been considered as medicine in the treatment of central nervous system disorders to date. A less favorable benefit-to-risk ratio might be justified in certain situations, such as in the treatment of epilepsy or dementia. Nevertheless, the necessity for more clinical trials is felt to clarify the effectiveness and eliminate the safety concerns, resulting in the development of new drugs based on DAs with lower toxicity and higher functional activities.

## 7. Future Perspectives

Some of the DAs showed extreme affinity for α7 nAChR, and are potential candidates for developing new research drugs in the treatment of cognitive disorders. Several DAs have prospecting potential in the treatment of Alzheimer’s disease. It is due to inhibiting the AChE and BChE enzymes in similar potency as standard drugs. They are also considered in developing the anticonvulsant drugs by blocking voltage-gated Na^+^ channels, or as analgesic in alleviation of chronic pain.

## Data Availability

The data presented in this study are available in the article.

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
