# Peer review of "Neuropharmacological Potential of Diterpenoid Alkaloids"

_pharmaceuticals, 2023, doi:10.3390/ph16050747_

Round 1

Reviewer 1 Report

This manuscript provides an overview of the potential neuropharmacological benefits of diterpenoid alkaloids (DAs) found in Aconitum and Delphinium species. These compounds have positive effects on neurological disorders, including antiepileptic, analgesic, acetylcholinesterase inhibitory, neuroprotective, antidepressant, and anxiolytic effects. However, their high neurotoxicity limits their widespread utilization, and few new drugs have been developed from DAs in recent years, despite their potential benefits. The manuscript highlights specific mechanisms by which DAs can have neuropharmacological effects, such as desensitizing Na+ channels or binding to α7 nicotinic acetylcholine receptors. Overall, the manuscript is informative and well-written.

Here are a few minor suggestions for improvement:

In the sentence that begins "Moreover, acetylcholinesterase inhibitory...", it might be clearer to separate out the different effects that were confirmed rather than listing them all in one sentence.

The phrase "less favorable benefit-to-risk ratio might be acceptable" could be rephrased to be more straightforward. For example, you could say "a less favorable benefit-to-risk ratio might be justified in certain indications, such as the treatment of epilepsy or dementia."

Author Response

Response is uploaded.

Reviewer 2 Report

The purpose of this review is to summarize the CNS effects and biomolecular mechanism of action of DAs and plants containing DAs, primarily Aconitum and Delphinium species. In December 2021, a systematic search of the databases Pubmed, Scopus, and Web of Science was conducted.

Overall its well written article but:

1- Part "2. Pharmacological activities" should be part 3 and there should be a general Part 2 regarding the main information, chemistry, types, Structural diversity, classification, toxicity etc and  it should contain enough space.

2- "Several methods were emphasized to reduce the toxic effects of aconites in traditional Chinese medicine substantially based on the heating processes (boiling, cooking, baking, steaming, and autoclaving) resulting in the hydrolysis of diester DAs to less toxic monoesters and lipoalkaloids" Provide the references for this statement.

3: Nothing has been concluded in conclusion or in Abstract regarding the structural features of reported compounds as SAR is not easy job here but at least the most effective chemical moieties can be highlighted in abstract, discussion and also in conclusion.

4: In table 3, there is one column regarding invitro or invivo study but i thing all studies are invivo than its useless to make column,just one sentence in enough in text body.

5-Figure 2.  Inside the figure word "receptor is missing in "(b) Homomeric α7 nicotonic acetylcholine receptor".  All figures are original? or reproduced with permission?

Author Response

Response uploaded

Reviewer 3 Report

I read the review on Neuropharmacological Potential of Diterpenoid Alkaloids by Salehi et al. with interest. I think the authors discussed all the necessary details, so that this compilation meets all the criteria of Pharmaceuticals. I am not too familiar with these particular compounds, so I don't know if the authors missed important contributions.
I would like to recommend acceptance of this contribution for publication, it will certainly meet some interest in the natural product community.

Author Response

Response: Thanks for the approval.

Reviewer 4 Report

The manuscript “Neuropharmacological Potential of Diterpenoid Alkaloids” (manuscript ID: pharmaceuticals-2314070) is a review about diterpenoid alkaloids usually found in Ranunculaceae family, such as in some species of the genera Aconitum, Delphinium, and Consolida, and their activity in central nervous system.  The manuscript is well designed and the text is clearly write. The scientific aim of this manuscript is interesting because there is few works if this focus available in literature. I suggest this paper could be reconsidered for publication after major corrections:

1)     Line 1: It is not an “Article”. It’s a “Review”.

2)     Please, rewrite the “Abstract”. It is repetitive (line 25-26).

3)     Line 18-19: Are DAs found in all the species of the genus Aconitum and Delphinium? Or “….found predominantly in some species of the genera Aconitum and Delphinium….”

4)     Both genera belong to Ranunculaceae family?

5)     Line 44-45: ….”characteristic components of some genera of Ranunculaceae family….”

6)     Please, reformulate the title of all the tables and figures presented in the manuscript.

7)     If the species was not identified, you must to use sp. Please, review it in all the tables.

8)     In the figures, the chemical structures must be better separated. They are very agglutinated.

9)     In Table 2: Include the rat model references.

10)  Line 207: The name of the species, such as A. bulleyanum, must be in italic.

11)  In Table 3: “Aqueous extracts” and “8-O-cinnamoylneoline”.

12)  Line 285: Write the complete words (acetylcholine) before abbreviation.

13)  In the section “Conclusion” it must be clearly described that this conclusions are about a literature review. Include this information in the conclusions.

14)  References: they are not according to the format of this journal. Please, reformulate it.

Author Response

Response uploaded

Reviewer 5 Report

Arash Salehi and co-workers compiled a narrative review to gather information on the bioactivity of diterpene alkaloids targeting the Central Nervous System. The manuscript is interesting and includes relevant information for readers. However, some points should be addressed before being considered further.

1.                  Detailed scrutiny should be performed throughout the manuscript to look for some grammar, stylistic, and even typos issues.

2.                  The abstract should be improved since it is not well structured and organized.

3.                  Line 22: Specify that this review corresponds to a narrative review.

4.                  A brief explanation for readers of the global/general structural features of diterpene alkaloids must be provided in the introduction section or even in a separate section, expanding more accurately the information in lines 51-54.

5.                  Unify the format of the structures throughout the manuscript, particularly in Figure 5.

6.                  Line 96: A scheme summarizing the biological effects of DAs must be depicted for readers.

7.                  Line 97: Anticonvulsant instead of Anticonvulsive. Be consistent throughout the manuscript.

8.                  Line 372: Is there any structure-activity relationship, at least global, that can be deduced from the compiled structural and bioactivity information? This analysis and discussion would be very useful for readers and better if accompanied by a comprehensive scheme indicating the structural moieties that can apparently be important for the various CNS-related activities.

9.                  A separate section including outlook and future perspectives is missing, highly related to the title.

10.              Another section gathering the clinical trials of Das is missing.

11.              Conclusions are presented as a summary of compiled information, but it can be improved focused on the mechanistic point of view and critical deductions/conclusions from the compiled information.

Author Response

Response uploaded

Round 2

Reviewer 2 Report

Authors have revised the manuscript as advised.

Reviewer 4 Report

The authors modified and improved the manuscript. Now I consider that it is able to publication in this journal.